# Always the Best Fit: Adaptive Domain Gap Filling from Causal Perspective for Few-Shot Relation Extraction

**Ge Bai, Chenji Lu, Jiaxiang Geng, Shilong Li, Yidong Shi, Xiyan Liu, Ying Liu,**
**Zhang Zhang**, **Ruifang Liu**[†]
Beijing University of Posts and Telecommunications, China [*]

## Abstract

Cross-domain Relation Extraction aims to transfer knowledge from a source domain to a different target domain to address low-resource challenges. However, the semantic gap caused by data bias between domains is a major challenge, especially in few-shot scenarios. Previous work has mainly focused on transferring knowledge between domains through shared feature representations without analyzing the impact of each factor that may produce data bias based on the characteristics of each domain. This work takes a causal perspective and proposes a new framework **CausalGF**. By constructing a unified structural causal model, we estimate the causal effects of factors such as syntactic structure, label distribution, and entities on the outcome. CausalGF calculates the causal effects among the factors and adjusts them dynamically based on domain characteristics, enabling adaptive gap filling. Our experiments show that our approach better fills the domain gap, yielding significantly better results on the cross-domain few-shot relation extraction task.

## 1 Introduction

Relation Extraction (RE) is one of the key tasks of Natural Language Processing (NLP), which aims to identify the relations between given entities. RE models (Zhang et al., 2017; Yamada et al., 2020) have impressive performance through large-scale supervised learning based on BERT (Devlin et al., 2019) and LSTM (Hochreiter and Schmidhuber, 1997). However, collecting sufficient amounts of data for certain classes may be laborious in practice. Although finetuning prompt-based pre-trained language models (He et al., 2023; Liu et al., 2022) have shown superior performance in few-shot RE tasks, they have encountered challenges in dealing with cross-domain problems. The main issue is that variables such as labels, syntactic structure,

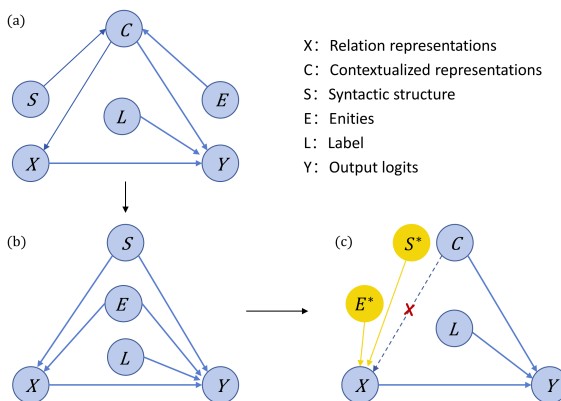

Figure 1: (a) An unified SCM for the task (b) Denoting variable $C$ by $S$ and $E$ (c) Blocking backdoor path by intervention and counterfactual generation

and entities having different distributions in each domain, resulting in data bias across different domains. From a causal perspective, the essence of this bias is that these variables have different causal effects on the results in different domains. Domain adaptation methods (Ganin et al., 2016; Shen et al., 2018) offer new insights to tackle these issues by transferring knowledge between domains through shared feature representations extracted from multiple domains. However, these work relies excessively on extracting shared features and label distributions while ignoring the unique feature of each domain, which would lead to inferior results when domains have significant semantic gaps. Therefore, Zhang and Lu (2022) proposed a label prompt dropout approach to eliminate the model's over-reliance on labels, but it is difficult to adequately capture the critical features of each domain by randomized dropout.

To address this issue, our work focuses on identifying and adjusting the causal effects of variables by considering the distinct characteristics of different domains. We propose a novel framework **CausalGF** and build a unified structural causal model (SCM) (Pearl et al., 2000) to describe the cross-domain RE task, as shown in Figure 1. The values and relationship in the graph can be altered

---
[‡]Ruifang Liu is the corresponding author.

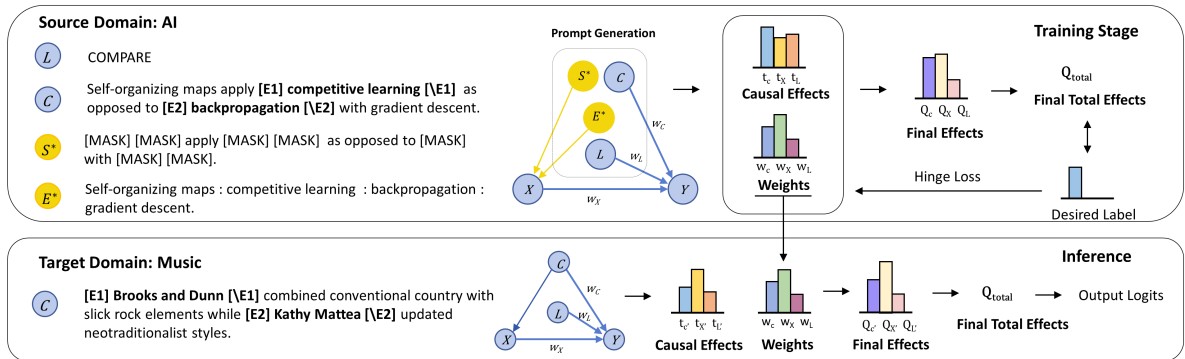

Figure 2: An overview of CausalGF. We utilize counterfactual to acquire representations for each variable and estimate their causal effects. The final effects are dynamically adjusted by intervention. Prompt generation, dynamic weighting, and loss function design are used to implement causal operations.

by intervention and counterfactual generation to study the causal effect of various factors. Furthermore, to adapt to the feature of different domains, it is necessary to estimate the causal effects of various factors and adjust them dynamically. CausalGF implements causal operations and dynamically adjusts the causal effects of variables through prompt generation, dynamic weighting, and loss function improvement, enabling adaptive gap filling based on domain characteristics.

We summarize the contributions as follows:

- To the best of our knowledge, CausalGF is the first work analyzing data bias and the influence of various factors from a causal perspective in cross-domain few-shot RE task.
- We dynamically estimate and adjust the causal effects of factors in training and inference, enabling adaptive gap filling according to the domain characteristics.
- Extensive experiments on different datasets and settings demonstrate the effectiveness of our approach. CausalGF outperforms previous state-of-the-art methods in all scenarios.

## 2 Methodology

The overall framework of CausalGF is shown in figure 2. Sections 2.1 and 2.2 describe the structural causal modeling and causal operation. 2.3 and 2.4 describe the implementation of the our method.

### 2.1 Structural Causal Modeling

As shown in Figure 1(a), cross-domain RE task is represented by unified SCM $\mathcal{G}$. The variable $C$ indicates the contextualized representations of an input text, which is output by the pretrain BERT encoder (Devlin et al., 2019). The variables $S$ and $E$ denote the syntactic structure and the representation of entities in the sentence respectively, which have a direct causal effect on $C$. Further, as there are no other parent nodes for node $C$, we can represent it by the nodes $S$ and $E$, as shown in Figure

1(b). $L$ denotes the label description. The variable $X$ is the representation of a relation for RE which is computed from $C$, and $Y$ indicates the output logits for prediction. On the edge $C \to X$, we fuse the semantic information and linguistic structures into SCM by adopting Transformer (Vaswani et al., 2017) to obtain the representation of node $X$. The causal effect of the parent node on $Y$ is obtained by full connectivity with a nonlinear transformation. The concepts and theories of causal inference are detailed in Appendix A.

### 2.2 Causal Operations

In our work, we aim to explore the causal effects of variables on the outcomes. From a causal perspective, that is, utilizing intervention and counterfactual generation as causal operations to explore the causal effects of the variables $S$, $E$, $L$, and $X$ on $Y$ in SCM $\mathcal{G}$. Since the variables $S$ and $E$ have causal effects on both $X$ and $Y$, their changes interfere with the calculation of the causal effect of $X \to Y$. Therefore, counterfactuals $S^*$ and $E^*$ are generated and intervened on $X$ upon to block the backdoor path (Morgan and Winship, 2014) and eliminate the causal effect of the original $S$ and $E$ on $X$ (Figure 1(c)). Meanwhile, the original $S$ and $E$ are preserved and restored to the original $C$ to estimate $C \to Y$ and maintain the semantic information of the original input.

For SCM $\mathcal{G}$, $X^*$ and $Y_{X^*}$, counterfactual of $X$ and the original prediction $Y_X$ are computed as:

$$X^* = X_{S^*, E^*} \tag{1}$$

$$Y_{X^*} = f_Y(do(X = X^*), S = S, E = E) \tag{2}$$

Where $f_Y$ is the function that computes $Y$.

### 2.3 Prompt Generation and Encoding

In order to implement the above causal operations, we obtained a representation of the above variables through prompt generation and encoding.

As shown in Figure 2, variable $L$ is represented by the label description. Counterfactual $E^*$ is obtained by extracting all entities in the sentence and connecting them in sequence. We mask these entities to obtain the counterfactual syntactic structure $S^*$. Besides, the original input $C$ is retained in prompt, and we add [CLS], [L], [S], and [E] as placeholder separators between variables. The given entity pair $\{e_{head}, e_{tail}\}$ are warped with special token [E1], [/E1], [E2] and [/E2] following the approach of Zhang et al. (2019).

$$T_{all} = [CLS] \, \mathbf{L} \, [L] \, \mathbf{C} \, [S] \, \mathbf{S}^* \, [E] \, \mathbf{E}_1^* \, ... \, \mathbf{E}_m^* \quad (3)$$

After prompt generation, the entire input instance $T_{all}$ is fed to the encoder:

$$h_L, h_{S^*}, h_{E^*} = Encoder(T_{all})_{L,S,E} \quad (4)$$

$$h_C = [Encoder(T_{all})_h, Encoder(T_{all})_t] \quad (5)$$

$$h_{X^*} = [h_{S^*}, h_{E^*}] \quad (6)$$

where $h$ is the output embedding for each token in $T_{all}$. $h_L$, $h_{S^*}$ and $h_{E^*}$ represent the respective representation of $L$, $S^*$ and $E^*$, which formed by final layer representations of the marker [L], [S] and [E], respectively. Base on SCM $\mathcal{G}$, $h_{X^*}$ is formed by concatenating $h_{S^*}$ and $h_{E^*}$. $h_C$ stands for the representation of the original sentence $C$ which is formed by concatenating the final layer representations of the entity markers [E1] and [E2].

## 2.4 Training and Inference

In the SCM $\mathcal{G}$, the parents of the outcome variable $Y$ are denoted as $\mathcal{E} = \{X, C, L\}$. During the training phase, we calculate the causal effect $t$ of each variable in $\mathcal{E}$ on $Y$ by utilizing the class prototype $r$ and variable representation $h$. To flexibly adjust the influence of each variable on the output, we introduce a learnable weight matrix $W = [w_{X^*}, w_C, w_L]$. The final causal effect $Q_{total}$ is computed using Formula (8).

$$t_i^k = r_k^\top h_i \quad (i = C, L, X^*) \quad (7)$$

$$
\begin{aligned}
Q_{total}^k &= \sum_{i \in \mathcal{E}} weight_i \cdot effect_i \\
&= \sum_{i=X^*,C,L} w_i \cdot t_i^k = \sum_{i=X^*,C,L} Q_i
\end{aligned}
\quad (8)
$$

The class prototype $r \in \mathbb{R}^{N_C \times H}$ is calculated by averaging the relation representations of the $N$ support instances of each class. Where $N_C$ indicates the number of classes, $H$ indicates the input hidden dimension. We optimize a hinge loss function by introducing the total casual effect $Q_{total}$.

$$\mathcal{L} = \frac{\sum_{k=1, k \neq y}^{N_C} max(0, m - Q_{total}^y + Q_{total}^k)}{N_C} \quad (9)$$

During inference, we choose the relation $y$ as the prediction by finding the closest class prototype to the query sentence's relation representation:

$$\hat{y} = \underset{k=0...N_C}{arg\,max} \, r_k^\top Q_{total}^q \quad (10)$$

Where $r_k$ is the class prototype of class $k$, $Q_{total}^q$ is representation of the query instance.

## 3 Experiment

### 3.1 Datasets and Implementation

We evaluate CausalGF on two cross-domain few-shot RE datasets: **CrossRE** (Bassignana and Plank, 2022): A manually-curated corpus contains 5265 sentences covering 6 domains with a unified label set of 17 relation types. To assess the domain adaptation of the model, we conducted experiments on CrossRE in single source domain and muitiple source domain scenarios. **FewRel**: FewRel 1.0 (Han et al., 2018) is collected from Wikipedia articles which contain 100 relations and 700 instances for each relation. FewRel 2.0 (Gao et al., 2019) contains test set from the biomedical contains domain 25 relations and 100 instances for each relation.

We compare CausalGF with the following baseline methods: **Proto-BERT** (Snell et al., 2017) is a prototypical network with BERT-base (Devlin et al., 2019) serving as the backbone. **BERT-PAIR** (Gao et al., 2019) is a method that measures the similarity of a sentence pair. **CP** (Peng et al., 2020) pretrains Proto-BERT using a contrastive pre-training approach that divides sentences into positive pairs and negative pairs. **HCRP** (Han et al., 2021) equips Proto-BERT with a hybrid attention module and a task adaptive focal loss. **Improved Domain Adaption (IDA)** (Yuan et al., 2022) proposes an encoder learned by optimizing a representation loss and an adversarial loss to extract the relation of sentences in the source and target domain. **LPD** (Zhang and Lu, 2022) introduces a label prompt dropout approach which is adaptable to cross-domain tasks.

We follow the pretraining method of LPD (Zhang and Lu, 2022) which pretrained on the Wikipedia dataset and on top of BERT-base from the Huggingface Transformer library. We perform multiple experiments with different random seeds and report the average accuracy together with the standard deviation. Detailed descriptions of experimental settings can be found in Appendix C.

### 3.2 Results and Analysis

**Main Results:** Table 1 shows that CausalGF outperforms all baseline models in CrossRE, achieving an average improvement of at least 1.90% and

| Models | 5-way-1-shot | | 5-way-5-shot | | 10-way-1-shot | | 10-way-5-shot | | Avg. | |
|---|---|---|---|---|---|---|---|---|---|---|
| | Multi | Single | Multi | Single | Multi | Single | Multi | Single | Multi | Single |
| Proto-Bert* | 67.70±0.5 | 52.2±0.7 | 80.71±1.0 | 64.65±0.8 | 58.65±0.9 | 39.86±1.2 | 76.82±1.1 | 50.82±00.8 | 70.97 | 51.83 |
| HCRP* | 70.47±1.0 | 60.34±0.9 | 85.05±0.3 | 70.68±1.5 | 59.17±0.5 | 48.53±0.6 | 78.51±1.0 | 60.70±0.9 | 73.30 | 60.06 |
| CP* | 78.33±0.9 | 49.96±0.7 | 86.89±1.1 | 70.70±1.2 | 70.95±1.1 | 44.45±0.9 | 78.36±1.4 | 53.82±0.7 | 78.63 | 54.73 |
| LPD* | 81.90±0.8 | 62.35±0.5 | 86.87±1.4 | 75.39±0.5 | 69.81±1.7 | 47.39±1.2 | 78.65±0.5 | 63.36±0.9 | 79.30 | 62.12 |
| **CausalGF** | **84.02**±0.7 | **63.88**±1.1 | **88.35**±0.3 | **76.44**±0.6 | **73.64**±0.4 | **49.57**±1.4 | **78.80**±0.8 | **64.93**±0.9 | **81.20** | **63.71** |

Table 1: Accuracy (%) of cross-domain few-shot classification on CrossRE music domain. (* These works have not been evaluated on CrossRE, so the results are produced by our implementation.)

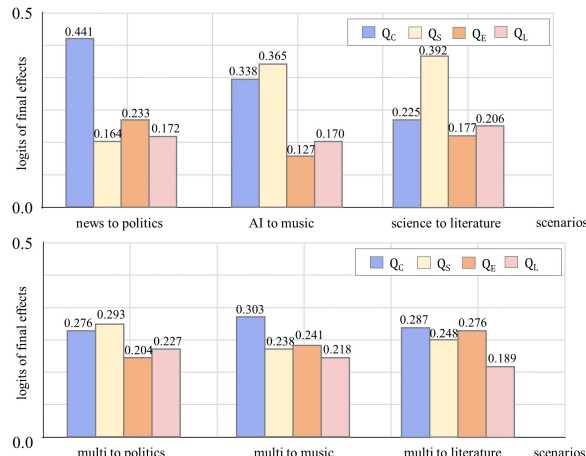

Figure 3: The final causal effects of the variable C, E, S and L in six different cross-domain scenarios.

1.59% for single and multiple source domain scenarios, respectively. Compared to the previous state-of-the-art LPD, our 10-way-1-shot results show enhancements of 3.83% and 2.18% with single and multiple sources, respectively. This highlights that adjusting causal effects adaptively is superior to random strategies in reducing the model over-dependence on labeling and data distribution. As shown in Table 2, our approach significantly outperforms HCRP, IDA, and CP by at least 4.44% and 4.10% in 1-shot and 5-shot settings, respectively. This indicates that learning the influence of variables based on causal theory is more effective than previous adaptive and contrastive methods in cross-domain few-shot tasks.

Experimental results for other domains can be found in Appendix D.

**Ablation Studies:** We construct ablation experiments on the music domain of CrossRE and Felrel2.0 dataset to investigate the contribution of each component in our approach. We implement a w/o counterfactual generation experiment by removing prompt generation, a w/o causal effect estimation by removing the weight matrix $W$, and a w/o causal effect adjustment experiment by initializing fixed $W$. Table 3 and 4 indicate that removing any part of our approach leads to varying degrees of decline in model performance. Notably, the results of

| Model | 5-way 1-shot | 5-way 5-shot | 10-way 1-shot | 10-way 5-shot |
|---|---|---|---|---|
| Proto-Bert | 40.12 | 51.50 | 26.45 | 36.93 |
| BERT-PAIR | 67.41 | 78.57 | 54.89 | 66.85 |
| HCRP | 76.34 | 83.03 | 63.77 | 72.94 |
| IDA | 76.30 | 84.71 | 67.87 | 75.84 |
| CP | 79.70 | 84.90 | 68.10 | 79.80 |
| LPD | 82.81±0.5 | 88.98±1.4 | 70.51±1.5 | 78.76±1.6 |
| CausalGF | **84.14**±0.9 | **91.10**±1.5 | **72.90**±1.1 | **83.92**±1.3 |

Table 2: Accuracy (%) of cross-domain few-shot classification on the FewRel2.0 test set.

| Model | muti source | single source |
|---|---|---|
| CausalGF | **81.20** | **63.45** |
| w/o counterfactual generation | 75.63 | 59.59 |
| w/o causal effects estimation | 77.17 | 61.71 |
| w/o causal effects adjustment | 75.33 | 60.29 |

Table 3: Ablation study results (%) of our methods on CrossRE dataset.

| Model | 5-way-1-shot | 10-way-1-shot |
|---|---|---|
| CausalGF | **84.14** | **72.90** |
| w/o counterfactual generation | 79.77 | 68.69 |
| w/o causal effects estimation | 81.10 | 71.35 |
| w/o causal effects adjustment | 80.65 | 70.89 |

Table 4: Ablation study results (%) of our methods on Fewrel2.0 dataset.

the w/o causal effect adjustment experiment reveal that improper utilization of causal effects can be counterproductive to the model's performance.

### 3.3 Causal Effects Across Different Domains

To verify the ability of our model adaptively filling gaps according to different domain characteristics, we explored the causal effect of each variable ($L$, $S$, $E$, $C$) on the results in different cross-domain tasks. Figure 3 shows the normalized average prediction logits for the ground truth which obtain from final causal effects $Qi$ ($i = L, S, E, C$) under six different cross-domain scenarios. The final causal effects of the same variables differ in different cross-domain scenarios, demonstrating that CausalGF adaptively fills gap by adjusting for causal effects.

For instance, in domains with significant differences, entities are less influential while syntax structure plays a more critical role since entities

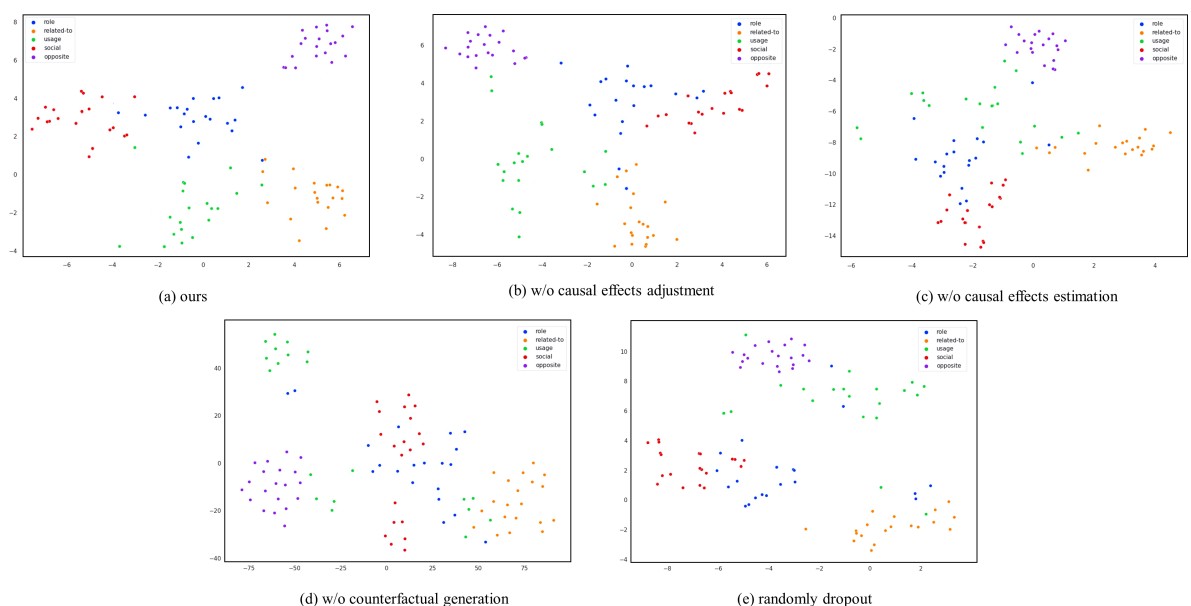

(a) ours

(b) w/o causal effects adjustment

(c) w/o causal effects estimation

(d) w/o counterfactual generation

(e) randomly dropout

Figure 4: The t-SNE visualization results for 100 samples from 5 labels in a 5-way-1-shot scenario on CrossRE.

| Model | 5-way 1-shot | 5-way 5-shot | 10-way 1-shot | 10-way 5-shot |
|---|---|---|---|---|
| Proto-Bert | 89.13 | 94.38 | 82.77 | 90.05 |
| BERT-PAIR | 88.32 | 93.22 | 80.63 | 87.02 |
| HCRP | 96.42 | 97.96 | 93.97 | 96.46 |
| CP | 95.10 | 97.10 | 91.20 | 94.70 |
| LPD | 98.17±0.0 | 98.29±0.2 | 96.66±0.0 | 96.75±0.2 |
| **CausalGF** | **98.31**1±0.3 | **98.54**±0.5 | **97.15**±0.3 | **97.04**±0.2 |

Table 5: Accuracy (%) of in-domain few-shot classification on the FewRel1.0 test set.

| Model | 5-way 1-shot | 5-way 5-shot | 10-way 1-shot | 10-way 5-shot |
|---|---|---|---|---|
| Proto-Bert* | 59.69±0.5 | 73.65±0.9 | 59.69±0.3 | 65.04±0.5 |
| HCRP* | 65.34±0.8 | 79.39±1.1 | 56.58±0.5 | 66.86±0.6 |
| CP* | 69.71±1.1 | 80.76±1.0 | 59.24±0.9 | 72.18±0.7 |
| LPD* | 76.62±1.0 | 79.70±1.2 | 69.14±1.2 | 70.69±1.5 |
| **CausalGF** | **78.51**±1.3 | **82.49**±1.0 | **70.90**±1.1 | **72.19**±0.8 |

Table 6: Accuracy (%) of in-domain few-shot classification on the CrossRE test set. (* are produced by our implementation.)

are domain-specific and semantic information of them is difficult for the target domain to utilize. In contrast, syntactic structure is more universal and applicable across various domains. This phenomenon is consistent with our intuition. We visualize the feature space to further demonstrate the effectiveness of CausalGF in Appendix **??**.

### 3.4 In-Domain Experiments

To further demonstrate the effectiveness of our method, we conducted additional experiments on in-domain tasks. We conducted experiments on Fewrel 1.0 and CrossRE datasets, and the results are shown in Table 5 and 6, respectively. The results demonstrate that our method achieves competitive performance in in-domain tasks. This in-

dicates that CausalGF has a universal capability to enhance the model's ability to learn features and make accurate predictions. It also proves the general significance of the causal effect estimation method in relation extraction tasks.

### 3.5 Visualization

During the process of model forwarding, we collect the vector representations of the test samples along with their respective category labels. The t-SNE toolkit (Van der Maaten and Hinton, 2008) is used to map the high-dimensional feature space of the test samples onto a two-dimensional plane, allowing for the measurement of sample similarity based on these representations. To evaluate the effectiveness of CausalGF, we implemented three ablation experiments and a random dropout method (LPD) to compare with our approach. Figure 4 shows the visualization results for 100 samples from 5 labels in a 5-way-1-shot scenario. The results clearly show that CausalGF outperforms other methods in terms of classification effectiveness. This signifies that the inclusion of counterfactual generation and the adaptation of causal effects greatly enhance the ability to learn domain-specific features.

### 4 Conclusion

In this paper, we propose CausalGF, a novel framework based on a causal perspective. By building a unified structural causal model, CausalGF estimates the causal effects of factors contributing to data bias and dynamically adjusts them to accommodate domain characteristics. Our model effectively fills the domain gap, outperforming strong baselines in various cross-domain scenarios.

## Limitations

Some limitations exist in our work. Our effectiveness is only examined on the task of relation extraction, while whether this method is able to generalize to other information extraction tasks, such as named entity recognition (NER) and event detection (ED), is not yet explored in this paper. In addition, a more fine-grained partition of variables with causal effects on the outcome may enhance the efficacy of counterfactual generation. The above issues will be explored in our future studies.

## Ethics Statement

Our contribution in this work is fully methodological, namely a novel framework from a causal perspective (CausalGF) to boost the performance of the cross-domain few-shot RE. Hence, there are no direct negative social impacts of this contribution.

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

## A Theory of Causal Inference

In this section we present the theory of causal reference involved in our approach.

**Structural Causal Model (SCM)**: Figure 5(a) shows an example of SCM. In general, the structural causal model ($G = \{V, U, F\}$) consists of two sets of variables and a set of functions (Glymour et al., 2016). The variable set $V = \{V_1 \ldots V_n\}$ represents the endogenous variables, which are the variables with observable causes (e.g. node $X$ and $Y$). The variable set $U = \{U_1 \ldots U_n\}$ represents the exogenous variables, which usually do not

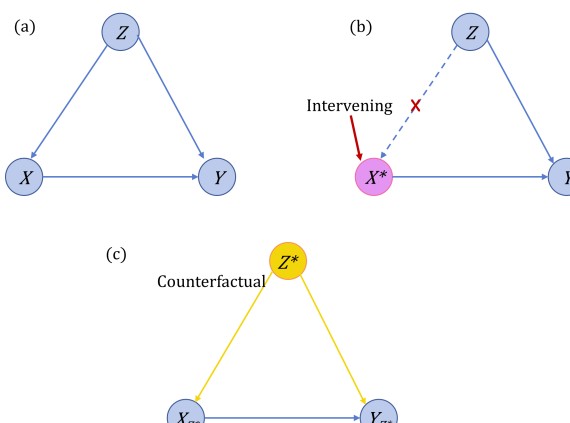

Figure 5: Example of a structural causal model. (a) Structural causal modeling (b) Intervening on variable $X$ (c) Counterfactual generation for variable $Z$

| Target Domain | Single Source | Multiple Source |
|---|---|---|
| Music | AI | Domains w/o Music |
| AI | Music | Domains w/o AI |
| Literature | Science | Domains w/o Literature |
| Science | Literature | Domains w/o Science |
| Politics | News | Domains w/o Politics |
| News | Politics | Domains w/o News |

Table 7: Domain segmentation of single source and multiple source on CrossRE dataset .

have parent nodes and whose causes are not usually taken into consideration (e.g. node $Z$). The function set $F$ is defined as $F = \{f_1, ..., f_n\}$, where $f_i$ represents the corresponding relationship between variables. The variables $V_i$ are determined by the functions as $V_i = f_i(A_i, U_i)$, where $A_i$ and $U_i$ represent the endogenous and exogenous variables, respectively, which have a direct causal effect on $V_i$. In terms of causality, the parent node is the cause and the children are the effect. As depicted in Figure 5(a), variable $X$ has a direct causal effect on variable $Y$ ($X \rightarrow Y$), while variable $Z$ has a direct causal effect ($Z \rightarrow Y$) and an indirect causal effect ($Z \rightarrow X \rightarrow Y$) on $Y$. The structural causal model provides a framework for comprehending the causal relationships between variables, allowing us to conduct experiments, make predictions, and intervene in these relationships.

**Intervening**: Intervening on a variable in a structural causal model involves fixing the value of that variable in order to study its correlation with other variables and its causal effects. Intervention can be represented by the do-calculus. For instance, the intervention on variable $X$ can be denoted as $do(X = x^*)$, where $x^*$ represents the given value (Pearl, 2009). As shown in Figure 5(b), after intervening on variable $X$, the causal relationship be-

tween $X$ and its parents will be cut off. Meanwhile, the backdoor path from $X$ to $Y$ ($X \leftarrow Z \rightarrow Y$) will also be blocked. At this point, variable $Z$ no longer has a simultaneous causal effect on $X$ and $Y$. Therefore, intervention can remove the confusion between variables and facilitate the estimation of the causal effect between variables.

**Counterfactual**: Counterfactuals emphasize the outcome of a hypothetical condition if a variable is hypothesized under identical conditions of reality. The concept of a counterfactual reflects a hypothetical scenario of "what would the outcome be if the variables were different". Unlike interventions which examine the effects on outcomes of implementing certain dispositional observations on variables in reality, counterfactuals focus on fictional scenarios that did not occur. As shown in Figure 5(c), assuming that the variable $Z$ is $Z^*$ in the case, the estimate of the causal effect on $X$ can be expressed as $X_{Z^*}$.

## B  Related Work

**Cross-Domain Few-Shot Learning:** In cross-domain few-shot learning, base and novel classes are both drawn from different domains, and the class label sets are disjoint. Although the supervised paradigm is effective in fundamental tasks, it suffers from the limitation of insufficient labeled data. To address this issue, previous work has proposed a variety of methods for Few-shot learning as well as domain adaptation.

Data-based few-shot learning methods augment the data with prior knowledge to overcome the difficulty of insufficient data (Gao et al., 2018; Wu et al., 2018; Cong et al., 2021). Algorithm-based methods leverage prior knowledge to search for an initial solution that is effective for multiple tasks simultaneously, which makes it facilitating the adaptation to new tasks (Finn et al., 2017; Yoo et al., 2018). Metric-based methods employ an encoder based on a metric to refine the sentence embedding in the latent space, allowing the learned latent space to generalize to novel relations with few labeled samples in the same domain (Triantafillou et al., 2017; Baldini Soares et al., 2019).

Domain adaption studies how to benefit from different but related domains. Shen et al. (2018) introduced Wasserstein distance to improve the generalization ability by constructing domain-invariant space between the source and target domain. Shi et al. (2018) employed an adversarial paradigm to extract class agnostic features in different domains.

| Models | 5-way-1-shot | | 5-way-5-shot | | 10-way-1-shot | | 10-way-5-shot | | Avg. | |
|---|---|---|---|---|---|---|---|---|---|---|
| | Multi | Single | Multi | Single | Multi | Single | Multi | Single | Multi | Single |
| Proto-Bert* | 52.2±0.7 | 48.88±1.0 | 64.65±0.8 | 60.04±0.5 | 39.86±1.2 | 38.66±0.8 | 50.82±0.8 | 47.78±1.1 | 51.88 | 48.84 |
| HCRP* | 60.34±0.9 | 58.05±0.6 | 70.68±1.5 | 69.82±0.5 | 48.53±0.6 | 45.07±1.1 | 60.70±0.9 | 54.69±0.3 | 60.06 | 56.91 |
| CP* | 62.58±0.7 | 43.46±0.9 | 69.82±1.2 | 69.68±0.6 | 51.99±0.9 | 41.63±0.7 | 62.37±0.7 | 58.35±0.3 | 61.69 | 53.28 |
| LPD* | 76.51±0.5 | 59.44±0.3 | 69.85±0.5 | 72.00±0.5 | 68.44±1.2 | 48.97±0.8 | 68.99±0.9 | 60.54±0.5 | 70.95 | 60.24 |
| **CausalGF** | **77.87**±1.1 | **61.34**±0.9 | **72.43**±0.6 | 67.90±0.3 | **69.20**±1.4 | **50.65**±0.8 | **71.61**±0.9 | **61.71**±1.0 | **72.78** | **60.40** |

Table 8: Accuracy (%) of cross-domain few-shot classification on CrossRE AI domain. (* These works have not been evaluated on CrossRE, so the results are produced by our implementation.)

| Models | Music | | AI | | Literature | | Science | | News | | Politics | |
|---|---|---|---|---|---|---|---|---|---|---|---|---|
| | Multi | Single | Multi | Single | Multi | Single | Multi | Single | Multi | Single | Multi | Single |
| Proto-Bert* | 70.97 | 51.83 | 51.88 | 48.84 | 66.78 | 54.28 | 67.16 | 67.53 | 67.22 | 62.86 | 66.20 | 44.69 |
| HCRP* | 73.30 | 60.06 | 60.06 | 56.91 | 70.38 | 62.45 | 68.72 | 63.97 | 68.29 | 61.41 | 66.70 | 53.76 |
| CP* | 78.63 | 54.73 | 61.69 | 53.28 | 75.93 | 55.97 | 69.77 | 56.51 | 61.65 | 62.85 | 72.80 | 48.40 |
| LPD* | 79.30 | 62.12 | 70.95 | 60.24 | 83.28 | 67.53 | 75.55 | 68.42 | 70.12 | 62.86 | 79.09 | 54.50 |
| **CausalGF** | **81.20** | **63.71** | **72.78** | **60.40** | **84.15** | **67.75** | **77.43** | **69.60** | **73.58** | **65.88** | **80.02** | **55.38** |

Table 9: Average accuracy (%) of cross-domain few-shot classification on CrossRE each domain. (* These works have not been evaluated on CrossRE, so the results are produced by our implementation.)

Yuan et al. (2022) proposed an encoder to extract the relation of sentences in the source and target domain and an adversary loss to merge the source domain and target domain.

Different from the above works that are based on conventional approaches, we tackle the cross-domain few-shot RE problem from the perspective of causal inference.

**Causal Inference:** Causal inference offers new insights for addressing the problem of data bias. In computer vision tasks, previous work has addressed the problem of unbalanced data distribution by causal manipulation of images through detection or segmentation methods (Tang et al., 2020; Abbasnejad et al., 2020). In NLP tasks, recent causal models have been applied in various tasks, such as text generation (Wu et al., 2020) and language understanding (Feng et al., 2021). Zeng et al. (2020); Wang and Culotta (2021) generated counterfactuals for weakly-supervised namely entities recognition (NER) and text classifications by replacing the target entity with another entity or their antonyms, respectively. Nan et al. (2021) mitigating the spurious correlations in long-tailed label distribution for information extraction tasks by counterfactual generation.

Unlike the previous methods, our approach utilizes causal operations to bridge the domain gap. We not only generate counterfactuals for cross-domain few-shot RE task, but also estimate and dynamically adjust causal effects to better align with the characteristics of each domain.

## C Implementation Details

We implemented our model with PyTorch 1.8.1. We use the Adam optimizer and set maximum length = 128, learning rate = 2e-5, batch size = 4, max iteration = 20,000 for CrossRE and FewRel 2.0. The learnable weight matrix $W = [w_{X*}, w_C, w_L]$ is initialized to $[0.5, 1, 1]$, and the learning rate for weight matrix = 5e-5.

During inference, we randomly sample 1,000 episodes from the N-way-K-shot support set and a query instance to evaluate our model. Following previous works (Han et al., 2018; Gao et al., 2019), we set N to 5 and 10, and K to 1 and 5. For all the experiments, we train and test our model on the 3090Ti GPU. It takes an average of 2.5 hours to run on the training dataset. All experiments are repeated five times with different random seeds under the same settings.

For Fewrel dataset, we follow the official split to use 64 relations of Fewrel 1.0 for training, 16 for validation and use FewRel 2.0 for testing to evaluate the domain adaptation of few-shot models. For CrossRE dataset, we construct our experiment with two scenarios: single source domain and multiple source domain. Domain segmentation on CrossRE dataset is shown in Table 7. For single source experiments, we select two different domains for source and tar, respectively. For multiple source experiments, we adopt the leave-one-out strategy.

| Label | Instance | Prediction of LPD | Prediction of CausalGF | Causal Effects |
|---|---|---|---|---|
| opposite | the **Kingdom of Judah** rebelled against the **Neo-Babylonian Empire** and was destroyed | win-defeat 0.427 compare 0.233 **opposite** 0.177 . . . . . . | **opposite** 0.720 role 0.101 win-defeat 0.087 . . . . . . | $Q_C$ 0.454 $Q_S$ 0.212 $Q_E$ 0.079 $Q_L$ 0.225 |
| win-defeat | **United Kingdom** lacks the charismatic leader needed to keep the country together and **Nazi Germany** successfully conquers Great Britain via Operation Sea Lion in 1940. | opposite 0.335 **win-defeat** 0.306 cause 0.197 . . . . . . | **win-defeat** 0.681 opposite 0.209 named 0.091 . . . . . . | $Q_C$ 0.361 $Q_S$ 0.104 $Q_E$ 0.327 $Q_L$ 0.178 |

Table 10: Two cases from CrossRE dataset, their source domain is news and target domain is politics.

# D Supplementary Experiments on CrossRE

we conducted 12 cross-domain experiments in 6 domains on the CrossRE dataset, as shown in the table 7. Except for the Music domain shown in section 3.2, we present the results of the remaining experiments here. For the AI domain, we present the detailed results of each few-shot setting in Table 8. For the other 4 domains, we report the average results of their experiments in Table 9. Specifically, CausalGF achieves an average improvement of at least 3.42% and 3.02% for single and multiple source domain scenarios, respectively. Furthermore, our approach achieves the best performance in all of the aforementioned cross-domain experiments, demonstrating that CausalGF adapts to domain characteristics and fills the gap in different cross-domain scenarios.

# E Case Study

As shown in Table 10, we show the content of two cases in the CrossRE dataset, and the prediction results of the previous sota LPD and our method. Since the semantics and application scenarios of the two labels (win-defeat and opposite) are relatively similar, LPD, a method that focuses mainly on the semantics of the labels, tends to confuse them, resulting in prediction errors.

Our approach can adaptively fill the domain gap by recognizing and dynamically adjusting the causal effects of different variables in a sentence. As a result, causalGF will not be overly dependent on a single variable in a sentence or be negatively affected by the similarity of a single variable in an instance. The table above shows the adjustment of causal effects by our method. It can be found that after adjustment, different variables have different causal effects, and our model can make correct and clear predictions in this way.