# OpenReview forum: "Always the Best Fit: Adaptive Domain Gap Filling from Causal Perspective for Few-Shot Relation Extraction"
_EMNLP/2023/Conference — EMNLP 2023 Findings_

### Official Review · Reviewer_8uUD · 2023-07-31

**Soundness:** 4

**Excitement:**

4: Strong: This paper deepens the understanding of some phenomenon or lowers the barriers to an existing research direction.

**Paper Topic And Main Contributions:**

This paper focuses on the cross-domain few-shot RE task from causal perspective. The authors propose a novel framework CausalGF based on structural causal model. The structural graph provides a theoretical explanation of feature selection and fusion for classification. Experimental results show that the proposed model outperforms baseline models on two benchmark datasets of this task.

**Questions For The Authors:**

1. In Figure 1, the edge from C to X is blocked and the counterfactual S* and E* are connected to X. However, it’s easy to observe another alternative, which blocks the edge from C to Y and connects S* and E* to Y. The resulting graph will be like S*, E*, L, X connected to Y and C connected to X. In the prompt encoding step, h_S and h_E will be separately weighted and contribute to the total causal effect, which is different to the proposed method. Here’s my question: why are you choosing the graph described in the paper instead of the above alternative? Is concatenating h_S and h_E better than separating them? If yes, would you please explain the reason?
2. Is this approach capable of migrating to regular cross-domain RE task?
3. In Figure 2, training on source domain is using S* and E*, but they are not used in inference. I don’t think S* and E* include label information, so would you please explain the reason or point out if I am missing some information?

**Reasons To Accept:**

This paper proposes a novel theoretical framework for classification feature selection and fusion, and experimental results demonstrate the effectiveness of the proposed method.

**Reasons To Reject:**

The motivation of graph topology selection is not well presented. See details in the first question in “Question For The Authors”.

**Reproducibility:**

3: Could reproduce the results with some difficulty. The settings of parameters are underspecified or subjectively determined; the training/evaluation data are not widely available.

**Reviewer Confidence:**

3: Pretty sure, but there's a chance I missed something. Although I have a good feel for this area in general, I did not carefully check the paper's details, e.g., the math, experimental design, or novelty.

---

> ### Author Rebuttal · Authors · 2023-08-27
>
> Thank you very much for the valuable and detailed review of our work. We will address your concerns below and make improvements based on your review.
>
> 1. We choose to intervene on X with S* and E* instead of intervening on Y because we want to preserve the direct causal effect of the original input text C on the output logits Y. The syntactic structure S and entity E are original parts that can mutually affect each other in the original input C, for example, the semantics of the same entity may vary in different contexts. Therefore, we choose to intervene on X, which not only preserves the direct causal effect of the original text but also generates counterfactuals for entities and syntactic structures and considers the causal effects of these two variables separately. We think the idea you propose is very interesting and we will do further experiments next to compare it with our design as well.
>
>     We concatenate h_S and h_E because S and E are two parts extracted from the original sentence C, and in real-world scenarios, these two parts exist as a whole. Therefore, our approach aims to use counterfactual generation during training and enable the model to learn the causal effect of syntactic structure and entities on the outcome. We do not perform counterfactual generation on S* and E* during inference. Instead, we expect the model to recognize the different causal effects of the syntactic structure and entities solely based on the original sentence and make accurate predictions. Besides, we have conducted a comparison of these two settings and found that concatenating h_S and h_E has a better performance. Thank you for the reminder, we will explain this phenomenon in the camera-ready version.
>
>    #
>
> 2. Our method is applicable in regular cross-domain relation extraction tasks and can yield competitive results. The main idea of our approach lies in identifying and adjusting the causal effects of different variables in order to fill the domain gap adaptively. Therefore, this causal operation can easily transferred to regular cross-domain RE tasks.
>
>     We apologize for not including experiments on regular cross-domain tasks in our submitted paper. This is because the main focus of our paper is how to effectively transfer knowledge between domains in the resource-limited few-shot scenario. Thank you very much for your suggestions. Due to time constraints, we are showing below the experimental results of the regular cross-domain RE task that we have already completed. We are still continuing to improve the design and analysis of the experiment, which will be showcased in the camera-ready version.
>
>
>
> | Model      | bc    | cts   | wl    | Avg.  |
> | ---------- | ----- | ----- | ----- | ----- |
> | GSN(2018) | 66.38 | 57.92 | 56.84 | 60.38 |
> | DRPC(2019)      | 67.30  | 64.28  | 60.19   | 63.92  |
> | MVC(2020)      | 70.32  | 66.43  | 64.61 | 67.12  |
> |   **CausalGF(ours)**    |   70.50    |    67.64   |    65.92   |    68.02   |
>
>
> The table above shows the F1 scores of the models on the ACE 2005 dataset over different target domains bc, cts, and wl. Our approach outperforms strong baselines and achieves competitive results.
>
>
> Dataset:
> **ACE 2005**[1] dataset include 6 different domains i.e., (bc, bn, cts, nw, un, and wl), covering text from news, conversations and web blogs.
>
> Baselines:
> **GSN**[2] is an adversarial learning model and is trained to learn the genre agnostic features for cross-domain RE.
> **DRPC**[3] is a deep structure-based model which employs dependency trees either as the input graphs to form the computation flow for deep learning models.
> **MVC**[4] simultaneously induces the structures and predicts the relations for the input sentences to obtain better domain adaptation.
>
> [1] Mo Yu, Matthew R Gormley, and Mark Dredze. Combining word embeddings and feature embeddings for fine-grained relation extraction. In NAACL-HLT, 2015.
>
> [2] Ge Shi, Chong Feng, Lifu Huang, Boliang Zhang, Heng Ji, Lejian Liao, and Heyan Huang. Genre separation network with adversarial training for cross genre relation extraction. In EMNLP, 2018.
>
> [3] Veyseh, A.P.B.; Nguyen,T.H.;and Dou, D. 2019. Improving cross-domain performance for relation extraction via dependency prediction and information flow control. In IJCAI.
>
> [4] Veyseh, Amir, et al. "Multi-view consistency for relation extraction via mutual information and structure prediction." Proceedings of the AAAI Conference on Artificial Intelligence.
>
>
> 3. We're sorry that the descriptions in our article have confused you. S* and E* do not contain labeling information. The label information is contained in the variable L, which plays a role in relationship extraction through prompt generation and causal effect adjustment. We add prompts including label information in the support instance to add relevant information into the model, and the query instance to be recognized does not contain labels.
>     The reason we do not use S and E in the inference process is that our model learns the causal effect of syntactic structure(S) and entities(E) during training, and thus the model can make accurate predictions through dynamic causal adjustment in the inference process, even without counterfactual generation.

---

### Official Review · Reviewer_Tq2e · 2023-08-04

**Soundness:** 4

**Excitement:**

4: Strong: This paper deepens the understanding of some phenomenon or lowers the barriers to an existing research direction.

**Paper Topic And Main Contributions:**

Previous work has mainly focused on transferring knowledge between domains through shared feature representations without analyzing the impact of each factor that may produce databases based on the characteristics of each domain. In this paper, the authors proposed a new framework CausalGF which is the first work analyzing data bias and the influence of various factors from a causal perspective in cross-domain few-shot RE tasks. The CausalGF outperforms previous SOTA methods in all scenarios.


**Reasons To Accept:**

1. The symbols are clear and the content is correct and comparatively novel.
2. It is the first work analyzing data bias and the influence of various factors from a causal perspective in cross-domain few-shot RE tasks.

**Reasons To Reject:**

1. Some content in the appendix needs to be adjusted to the main text to improve clarity and conciseness. For example, consider moving a brief description of the baseline from Appendix D to Section 3.1.


**Reproducibility:**

5: Could easily reproduce the results.

**Reviewer Confidence:**

2: Willing to defend my evaluation, but it is fairly likely that I missed some details, didn't understand some central points, or can't be sure about the novelty of the work.

---

> ### Author Rebuttal · Authors · 2023-08-27
>
> Thank you very much for the comprehensive review of our work,  we will make improvements and refinements based on your review. We will move some important experimental results (such as Appendex D) and implementation details from the appendix to the main text in the camera-ready version.

---

### Official Review · Reviewer_igC2 · 2023-08-05

**Soundness:** 3

**Excitement:**

3: Ambivalent: It has merits (e.g., it reports state-of-the-art results, the idea is nice), but there are key weaknesses (e.g., it describes incremental work), and it can significantly benefit from another round of revision. However, I won't object to accepting it if my co-reviewers champion it.

**Paper Topic And Main Contributions:**

This paper focuses on the cross-domain few-shot Relation Extraction task. It proposes an unified structral causal model called CausalGF from the causal perspective. This model adaptively fills the data bias gap across diffrent domains, using intervention and counterfactual generation causal operation to achieve this purpose. Extensive experiments on different datasets and settings demonstrate the effectiveness of this approach.

**Questions For The Authors:**

Please refer to "Reasons To Reject".

**Reasons To Accept:**

(1) The causal perspective to mitigate the domain gap for cross-domain RE task is sound.

(2) The method is elaborated through theoretical proof, which is interesting.

(3) The method obtains significant improvements over multiple baseline models.

**Reasons To Reject:**

(1) Lack of necessary case study to demonstrate why this method works.

(2) The reason why syntactic structure and entities are distinct cross-domain gaps is not clarified.

(3) Detailed experiment settings should be provided such as hyperparameter and data preprocessing.

**Reproducibility:**

3: Could reproduce the results with some difficulty. The settings of parameters are underspecified or subjectively determined; the training/evaluation data are not widely available.

**Reviewer Confidence:**

4: Quite sure. I tried to check the important points carefully. It's unlikely, though conceivable, that I missed something that should affect my ratings.

---

> ### Author Rebuttal · Authors · 2023-08-27
>
> Thank you very much for the valuable and detailed review of our work. We will address your concerns below and make improvements based on your review.
> 1. We apologize for the omission of case study in our submission, and we will show two cases from CrossRE below to further demonstrate the effectiveness of our approach and we will add it to the camera-ready version.
>
> | Label  | Instance | Prediction of  baseline LPD | Prediction of   CausalGF  | Causal Effects  of Variables  |
> | ------ | ------- | -------- | --------- | -------- |
> | opposite        | the **Kingdom of Judah** rebelled against the  **Neo-Babylonian Empire** and was destroyed          | win-defeat  0.427   compare 0.233  **opposite** 0.177  ……   | **opposite**  0.720   role 0.101  win-defeat 0.087  ……    | Q_C 0.454,   Q_S 0.212,  Q_E 0.079,  Q_L 0.225   |
> | win-defeat     | **United Kingdom** lacks the charismatic leader   needed to keep the country together and **Nazi Germany** successfully conquers Great  Britain via Operation Sea Lion in 1940 .         | opposite  0.335   **win-defeat** 0.306  cause 0.197   ……    | **win-defeat**  0.681   opposite 0.209  named 0.091  ……    | Q_C 0.361,   Q_S 0.104,  Q_E 0.327,  Q_L 0.178   |
>
> We show the content of two cases in the CrossRE dataset (their source domain is news and target domain is politics), and the prediction results of the previous sota LPD and our method. Since the semantics and application scenarios of the two labels（win-defeat and opposite）are relatively similar, LPD, a method that focuses mainly on the semantics of the labels, tends to confuse them, resulting in prediction errors.
>
> Our approach can adaptively fill the domain gap by recognizing and dynamically adjusting the causal effects of different variables in a sentence. As a result, causalGF will not be overly dependent on a single variable in a sentence or be negatively affected by the similarity of a single variable in an instance. The table above shows the adjustment of causal effects by our method. It can be found that after adjustment, different variables have different causal effects, and our model can make correct and clear predictions in this way.
>
> 2. On the one hand, from an intuitive perspective, syntactic structure and entities have different characteristics and play different roles in relation extraction tasks. Specifically, entities in a sentence, especially the head and tail entities that need to be identified, contain word-level semantic information that is beneficial for relation extraction while syntactic structure contains more contextual information. However, in cross-domain tasks, the domain-specific entity information is more difficult to be utilized by the target domain while syntactic structure is more universally applicable. On the other hand, from a causal perspective, entities and syntactic structures have different causal effects on the results of cross-domain relation extraction, so it is necessary to separately identify and adjust these effects in the task. In section 3.3 of our paper, we demonstrate the distinct causal effects of different variables in different cross-domain tasks.
>
> 3. We're sorry that our typography has inconvenienced your reading. Due to space constraints, we have included implementation details such as hyperparameter tuning and data preprocessing in Appendix C. In the camera-ready version, we will move essential implementation details to the main text.

---

### Meta-Review · Area_Chair_egTs · 2023-09-19

**Recommendation:** 4

**Metareview:**

This paper explores a causal model for cross-domain few-shot relation extraction. The reviewers have several concerns for the paper regarding the lack of clarify and motivation for some model details. However, the rebuttal provides further information and seems to address the major concerns. The authors are encouraged to update the paper according to the discussion with the reviewers.

---

### Decision · Program_Chairs · 2023-10-07

**Decision:**

Accept-Findings

**Comment:**

This paper explores a causal model for cross-domain few-shot relation extraction. The reviewers have several concerns for the paper regarding the lack of clarify and motivation for some model details. However, the rebuttal provides further information and seems to address the major concerns. The authors are encouraged to update the paper according to the discussion with the reviewers.